# Effects of school menstrual hygiene management, water, sanitation, and hygiene interventions on girls' empowerment, health, and educational outcomes: Lasta district, Amhara regional state, Ethiopia

**Fisseha A. Andargie** [1]*, **Femi R. Tinuola**[2]

**1** Department of Public Health Administration, Central University of Nicaragua, Managua, Nicaragua,
**2** Department of Sociology, Federal University Gusau, Gusau, Zamfara, Nigeria

* fissehaatalie@yahoo.com

## Abstract

This study assessed the effects of school-based menstrual hygiene management (MHM) and water, sanitation, and hygiene (WASH) interventions on the empowerment, health, and educational outcomes of menstruating girls using cross-sectional and experimental designs. It examined whether access to MHM education and WASH facilities could enhance girls' self-confidence, physical, emotional, and social health, class attendance, and academic performance. The results showed significant improvements in empowerment at intervention schools, with 54% of girls feeling confident purchasing sanitary products, compared to 18% in control schools, indicating better emotional well-being. Additionally, 21% and 22% of girls in intervention schools felt comfortable discussing MHM with boys and mothers, respectively, compared to just 9% in control schools, reflecting improved social health. Regarding physical health, 51% of menstruating girls in intervention schools practiced genital hygiene three to four times a day, compared to 33% in control schools. Educational outcomes were also improved, with 68% of girls in intervention schools attending class during menstruation, compared to 30% in control schools, and 78% reporting adequate study time at home, compared to 41% in control schools. However, no significant difference in academic performance was found between the two groups. Overall, the findings suggest that school-based MHM and WASH interventions can significantly empower menstruating girls, improve their physical, emotional, and social health, and reduce menstrual-related absenteeism.

## 1. Introduction

Menstruation is a natural physiological process involving the monthly shedding of the uterine lining, occurring among pubescent girls and pre-menopausal women who are not pregnant [1,2]. Menstrual hygiene management (MHM) refers to the hygienic handling of menstrual blood using appropriate sanitary products and hygiene practices, including absorbent products, soap, water, and safe disposal of used materials [1,3].

**Data availability statement:** All relevant data are within the manuscript and its Supporting Information files.

**Funding:** The author(s) received no specific funding for this work.

**Competing interests:** I have read the Journal's policy and authors of this manuscript have declared that they have no competing interest.

Despite its importance, discussions on MHM were considered taboo especially in developing nations [4,5]. Interest in MHM emerged in the 2000s when private and non-governmental organizations began promoting MHM education to pubescent girls, primarily within school settings [6].

Therefore, menstruation and MHM is a relatively young field of study, having gained research attention over the last 25 years. In this brief period, research has primarily focused on examining the relationship between menstruation and girls' class absenteeism [7–9].

The United Nations Children's Fund (UNICEF), Plan International, Save the Children, and Emory University in the Philippines conducted an evaluation and identified three key determinants of MHM within school environments that could empower girls. These determinants include access to water, sanitation, and hygiene (WASH) facilities and services, MHM education and practical guidance on hygiene practice, and access to sanitary products. The evaluation reported that the absence of these determinants could lead to unsafe MHM practices, confusion, and anxiety [10].

In this study report, empowerment refers to an individual or group's increased control over their lives, characterized by self-confidence, access to adequate knowledge, and behaviour change, which are essential for making informed decisions regarding health and education [11].

Since 2012, UNICEF and the University of Colombia have hosted annual conferences under the theme "WASH in Schools Empowers Girls' Education." Proceedings from these conferences indicate that school WASH interventions have the potential to enhance girls' education and improve MHM practices [12].

Moreover, access to school WASH services can help prevent at least 9.1% of the global disease burden [13], and improvements in health outcomes can contribute to higher class attendance and better academic performance [14–16].

Conversely, the absence of school WASH services can negatively affect girls' emotional, social, and physical health, leading to confusion and anxiety due to fear of menstrual leakage, social isolation, and genital bruising from unsafe MHM practices [10]. The World Health Organization defines health as "A state of complete physical, mental, and social well-being, and not merely the absence of disease" [17].

Despite evidence of positive associations between school WASH services, girls' MHM practices, and class attendance, some studies have reported no significant correlation between MHM practices and class attendance.

Studies conducted in Nepal and Malawi, for instance, reported that the effect of menstruation on class attendance was minimal. Instead, they attributed class absenteeism to household responsibilities and lack of motivation [9,18]. These findings contrast with studies from northern Ethiopia and Kenya, which observed improved class attendance among girls following school WASH interventions [8,19].

However, there is broad agreement on the correlation between class attendance and academic performance.

A study found that consistent class attendance, along with regular assessments and study time, was positively correlated with academic performance [20]. In addition, a study on higher education reported that regular attendance reduces the likelihood of poor academic performance [21]. However, while class attendance benefits lower-performing students, this relationship does not extend to higher-performing students [22].

Generally, while studies on the association between MHM and class attendance have laid the groundwork for further research, they lack comprehensiveness [23].

First, evaluation reports from Philippine-based non-governmental organizations, UNICEF, and the University of Colombia were qualitative; they were not supported with quantitative findings [10,11]. Additionally, no studies, aside from qualitative project completion reports, have explored the empowering potential of school WASH interventions.

Second, the findings across studies lack consistency. For instance, research in Nepal and Malawi reported minimal effects of MHM on class attendance, in contrast to the northern Ethiopia and Kenya studies that found improved attendance following school WASH interventions, highlighting contradictions in the literature [8,9,18,19].

Third, existing research findings have been confined to in-school factors, overlooking external influences such as educational policies, community dynamics, and peer influence on girls' MHM practices [7–9,17]. However, in-school and out-of-school environments play critical roles in shaping these practices [24].

Finally, studies on the relations between girls' MHM practice and health have primarily focused on clinical aspects, such as reproductive tract infections, while neglecting key social, emotional, and physical health dimensions [25].

Therefore, the significance of this study stems from the limitations identified in previous studies on MHM and class attendance. Accordingly, the study aimed to examine the effects of school-based menstrual hygiene management, water, sanitation, and hygiene interventions on girls' empowerment, health, class attendance, and academic performance.

## 2. Materials and methods

### 2.1. Study setting and description of school interventions.

The study was conducted in Lasta District, North Wollo Zone, Amhara Regional State, located in the northern and northwest part of Ethiopia.

Lasta District Education Office identified 35 schools with critical WASH service shortages in 2017. Plan International Ethiopia, an international NGO, randomly selected 10 Primary 2nd Cycle schools from 35 schools for MHM and WASH interventions.

**Contents of the interventions:** The key components of the program included MHM education, the provision of sanitary products and soap, and access to water and gender-segregated toilets. Additionally, the intervention established MHM rooms, where girls could rest when feeling unwell, change sanitary pads during menstruation, prepare sanitary products, and participate in peer-to-peer MHM education.

In contrast, control schools lacked the facilities and hygiene services provided through the MHM and WASH interventions. Their only support came from the District Women and Children Affairs Office, which offered only MHM general awareness as part of the education policy.

While intervention schools received MHM education and WASH services, most other schools in the district, including control schools did not due to budget constraints and the absence of other WASH actors providing school WASH services. Therefore, the lack of access to WASH services in control schools was not a result of the study design but rather a consequence of district budget limitations and the absence of external WASH support.

The frequency of MHM education and WASH service support for menstruating girls varied depending on the type of service. Long-term facilities, such as improved latrines and MHM rooms were constructed for sustained use, while consumable items like sanitary pads and soap required regular replenishment.

The MHM education continued until all students in the intervention schools, including boys, received training. Afterward, school MHM club members held monthly meetings to monitor MHM practices, support pubescent students, and provide guidance. Additionally, the MHM club organized training sessions for new students at the beginning of each academic year.

A primary goal of the school interventions was to equip menstruating girls, schoolteachers, and the district education office with the skills needed to sustain the interventions'

achievements using their resources, gradually reducing and eliminating reliance on external support.

The school intervention spanned six years, from July 2018 to January 2024. However, activities were primarily conducted during school terms and were temporarily interrupted during school breaks, the COVID-19 pandemic, and district conflicts.

Throughout the intervention period, program facilitators from Plan International Ethiopia, in collaboration with the District Women and Children Affairs Office and schoolteachers, conducted regular follow-ups to ensure that female students acquired adequate MHM skills and to address service gaps as they arose.

To minimize the risk of contamination in control schools, the study used geographical isolation as its primary strategy. Both intervention and control schools were located in rural, mountainous areas where communities were sparsely populated and schools were far apart. Furthermore, due to concerns about gender-based violence and security, girls in rural communities typically did not travel long distances to attend school. As a result, only menstruating girls living near schools had the opportunity to continue their education. This geographical isolation effectively prevented the transfer of MHM knowledge from intervention schools to control schools.

Additionally, before selecting the control schools, the research team verified whether any knowledge transfer had occurred between intervention and control schools or if other WASH interventions were present in or near control schools.

Therefore, to address contamination concerns, the research team conducted the study in rural schools, confirmed the absence of similar interventions in control schools, and ensured that no deliberate attempt was made to transfer knowledge from intervention schools to control schools.

## 2.2. Study design

The study employed a cross-sectional design as the school interventions began in July 2018, while the study commenced in April 2023.This timed difference made a longitudinal design infeasible as it made it impossible to compare baseline data and intervention outcomes. Instead, the study compared intervention schools with control schools that had not been exposed to similar interventions.

Although the study used random sampling to select both schools and participants, neither the schools nor the girls could switch between groups. Girls from intervention schools had no opportunity to be part of the control group, and the same applied to girls from control schools. Therefore, the study followed a quasi-experimental design.

## 2.3. Methods and data collection instruments

The study employed qualitative and quantitative methods to address potential gaps associated with using a single approach. In terms of sequence, the qualitative method was applied first to adjust the quantitative method as needed. However, some focus group discussions and key informant interviews were conducted solely for qualitative data collection.

To collect qualitative data the study utilized focus group discussions, key informant interviews, and observations; whereas for quantitative data collection, questionnaires, health centre records and half-year academic performance data were used.

## 2.3. Study population

The study aimed to assess the effects of school interventions on menstruating female students. Therefore, only menstruating girls comprised the study population. However, to evaluate the

influence of communities, peers, and policies on girls' MHM practices, male students, mothers of girls, and district office heads were included for qualitative data collection.

### 2.4. Ethical issues

Participant informed assent: A day before participating in the study, the study team briefed the girls on its purpose. The following day, after having sufficient time to discuss their participation with parents and peers, the girls provided written assent by signing the designated space on the questionnaire.

Guardian informed assent: Two days before data collection, the study schools, with support from the Parents and Teachers' Association (PTA)—a body that facilitates collaboration between schools and communities on educational matters, invited girls' parents to a brief morning meeting. During this session, the study team provided an overview of the study, and following discussions, parents were asked whether they would allow their daughters to participate.

As a result, all parents provided verbal consent and those who participated in the focus group discussions gave written consent by signing the consent form.
Institutional written assent: Additionally, the study obtained written consent from Lasta District Education Office which was responsible for monitoring, coordinating, and managing all schools in the district. The full written ethics statement reads as follows:

Date: 23Feb.2023

Lasta District Education Office, North Wollo Zone of Amhara Region, understands the relevance of examining the *"Effects of School Menstrual Hygiene, Water, Sanitation and Hygiene Interventions on Girls Hygiene Knowledge and Practice, Empowerment, Health, and Educational Outcomes"* at the Primary 2nd Cycle Schools found in Lasta District.

The Office recognizes that the study will include menstruating girls, boys, teachers, and community representatives around the study schools and Heads of WASH Sector Offices in Lasta District. We believe that the results of the study may benefit for Lasta District Education Office in particular and Amhara Regional State in general to improve school WASH interventions.

Therefore, Lasta District Education Office has given this ethical clearance letter/permission to conduct the inquiry in the schools and kindly request the study schools and district WASH Sector Offices to support in providing the required data to Mr. Fisseha Atale, who is conducting the study.

### 2.5. Sampling method and sample size

The study employed a stratified sampling method to group intervention and non-intervention schools based on their similarity in access to MHM education and WASH services. The research team assigned codes to 10 intervention schools and then mixed these codes. From the mixed set, three school codes were randomly drawn. The same procedure applied to 25 control schools. As a result, three intervention schools were selected from the 10 WASH intervention schools, while three control schools were chosen from 25 control schools in the district.

To select girls the study followed a multi-stage sampling approach. First, male students were excluded since the study focused on female students. Second, non-menstruating girls were also excluded, as the study aimed to capture the experiences of menstruating girls.

After these exclusions, the research team identified 333 menstruating girls from intervention schools and 293 from control schools, yielding a total study population of 626. Adjusting for the probability of non-attendance and applying a sample size formula, the final sample size

was determined to be 360. Within this sample, 192 girls (53%) were allocated to intervention schools, while 168 girls (46.6%) to control schools, maintaining proportional representation.

Then the study team prepared a separate list of menstruating girls and schools received their sample allocation based on their proportional contribution to the study population. The lottery method was used to allocate samples for each school. For a school that had 50 menstruating girls on the list and a sample allocation of 25, the research team prepared 50 draws, 25 labelled from 1–25. The remaining was marked as zero. After mixing the codes, each girl, in the order of the list, drew a code. If the draw contained a number between 1 and 25, she was included in the final sample; if it was zero, she was excluded.

The research used a 95% confidence level and 5% margin of error, with an expected variance of 0.5. Given the known student population, the research applied a finite population formula, detailed as follows [26]:-

$$S = Z^2 NP(1-P)/e^2(N-1) + Z^2 P(1-P)$$

where
S: stands for sample size
N: Population size of students
P: Population proportion, with an expected variance of 0.5
Z: Z-score, which is 1.96 in this case;
e- Margin of error/degree of accuracy, which is 0.05 (5%);

For quality control, the study employed single blinding to minimize biases during data collection. Female students from intervention schools had been engaged in school interventions for several years, and their perspectives were influenced by their roles in implementing these interventions. Additionally, the confidence gained through these programs gave them experience to share their views.

For girls in control schools, the study team communicated clearly that the study was not associated with any future opportunities for school WASH interventions. They were informed to share their individual MHM practice and access to WASH services in the school environment.

For data collectors, the research team provided comprehensive training including how to avoid being biased during data collection.

To further control researcher bias and enhance data quality, the study adopted a mixed-method approach for data collection. Clinical data related to MHM were gathered, cross-checked against field data, and analysed using the independent samples t-test to ensure accuracy in the interpretation of differences between the two groups of menstruating girls. Furthermore, the study employed a socio-ecological perspective, to avoid confirmation bias, allowing for the analysis of multiple levels of factors.

## 2.6. Mitigating effects of extraneous variables

To minimize the effects of confounding variables on the study findings, the study used the following methods:-

Restriction: Most girls (69%) were between 13 and 15 years old and enrolled in Primary Second Cycle Schools (Grades 7 and 8). These restrictions helped establish comparable characteristics in age and education

Matching: Girls' parents, in intervention and control schools, reside in rural areas where 82% rely on agriculture for their livelihood. The study schools are also located within these rural communities. To control for potential differences in study findings due to variations in family background (urban vs. rural), income sources (farming vs. business), and school

location (rural vs. urban), the study was conducted in schools and among parents that had similar characteristics.

**Distance from school and academic performance** Potential differences in findings between the two groups of girls, due to variations in school distance, urban vs. rural settings, and the use of prior academic performance, were controlled by selecting schools in similar rural environments and using girls' average academic scores from one semester, recorded at the time of the study.

Teaching quality and learning environment: All teachers in the intervention and control schools have completed four years of university education in teaching. Additionally, since these schools are located far from urban areas, they have similar access to library resources and other educational technologies. Therefore, significant differences in teaching quality and access to educational technology between the intervention and control schools were unlikely.

**Cultural and religious homogeneity** The majority of menstruating girls in the study (98%) were from the Amhara ethnic group, characterising similar culture, and 84% identified as Orthodox Christians. This demographic homogeneity reduced the potential influence of confounding variables related to differences in ethnicity and religion.

## 2.7. Data collection

The study team provided data collectors' training, including pilot testing on March 7 and 8, 2024, respectively, followed by data collection that lasted from March 11–15, 2024.

## 2.8. Statistical methods used

The preparation for data analysis involved data cleaning, coding of qualitative and quantitative data, and entering the data into spreadsheets. The study used thematic analysis for qualitative data, while SPSS software was employed for quantitative analysis.

Descriptive statistics, including frequencies and percentages were used to summarize the characteristics of each question, and an independent samples t-test was conducted to evaluate statistical significance. Finally, findings from qualitative and quantitative analyses were integrated to assess the research hypotheses.

## 3. Results

### 3.1. Sample characteristics

The demographic analysis revealed that girls aged 13–15 made up the majority at 69%, followed by those aged 16–18 at 16% and 10–12 at 15%. In terms of ethnicity, 98% of participants identified as Amhara, with smaller proportions identifying as Agew (1%) and Tigre (0.8%). Regarding religion, 84% of the girls identified as Orthodox, 9% as Protestant, and 6% as Muslim.

The result indicated that 66% of fathers and 52% of mothers were literate. Moreover, 82% of parents depended on agriculture as their main source of income, while only 8% earned a living through business (Table 1).

### 3.2. Empowering girls through school interventions

**3.2.1. Qualitative results.** Girls from intervention schools reported having open discussions about menstrual health management (MHM) with their mothers, sisters, male peers, and teachers. A Girls' Club facilitator at one such school noted, *"Girls no longer hesitate to ask for menstrual breaks during class."*

**Table 1. Demographic characteristics of the sample.**

| N=360 | Characteristics | Frequency | Percent |
|---|---|---|---|
| Age | 10–12 | 55 | 15.3 |
| | 13–15 | 248 | 68.9 |
| | 16–18 | 56 | 15.6 |
| | Above 19 | 1 | 0.3 |
| Religion | Orthodox | 302 | 83.9 |
| | Islam | 22 | 6.1 |
| | Protestant | 34 | 9.4 |
| | Other | 2 | 0.6 |
| Ethnic group | Amhara | 353 | 98.1 |
| | Agew | 4 | 1.1 |
| | Tigre | 3 | 0.8 |
| Parents' occupation | Farmer | 294 | 81.7 |
| | Merchant | 30 | 8.3 |
| | Office worker | 16 | 4.4 |
| | Other | 20 | 5.6 |
| Mothers' level of education | Abel to read and write | 144 | 40 |
| | Grade 1–6 | 28 | 7.8 |
| | Grade 7–12 | 14 | 3.9 |
| | Not able to read and write | 172 | 47.8 |
| | Other | 2 | 0.6 |
| Fathers' level of education | Abel to read and write | 183 | 50.8 |
| | Grade 1–6 | 43 | 11.9 |
| | Grade 7–12 | 14 | 3.9 |
| | Not able to read and write | 118 | 32.8 |
| | Other | 2 | 0.6 |

In contrast, girls in control schools stated that their discussions about MHM were mostly limited to female teachers, as they felt uncomfortable discussing the topic with boys and male teachers. A female teacher at a control school observed, *"Girls lack sufficient knowledge about MHM and often do not have the confidence to discuss the topic with their teachers and peers."*

**3.2.2. Quantitative results.** The data revealed that 54% of girls in intervention schools felt confident purchasing sanitary products, compared to only 18% in control schools. Meanwhile, 31% of girls in intervention schools reported feeling shy when buying sanitary products, whereas this was the case for 58% of girls in control schools (Fig 1 and S2 Fig).

The quantitative data on girls' social interactions revealed that 21% and 22% of girls from intervention schools reported discussing MHM with boys, parents, and others, compared to only 9% of girls in control schools. In terms of access to MHM education, 33% of girls in control schools stated they had not received any MHM education, whereas this figure was just 2% in intervention schools (Fig 2).

The Independent Samples T-test indicated that the P-values for differences in girls' social interactions between the two groups ranged from 0.001 to 0.022. Thus, all values fall below the significance level of 0.05 (P <α) (Table 2).

## 3.3. Effects of school interventions on girls' health

Shumshiha, one of the district health centres, provides healthcare services to the community, including female students. Clinical data that covered from September 2022 to February

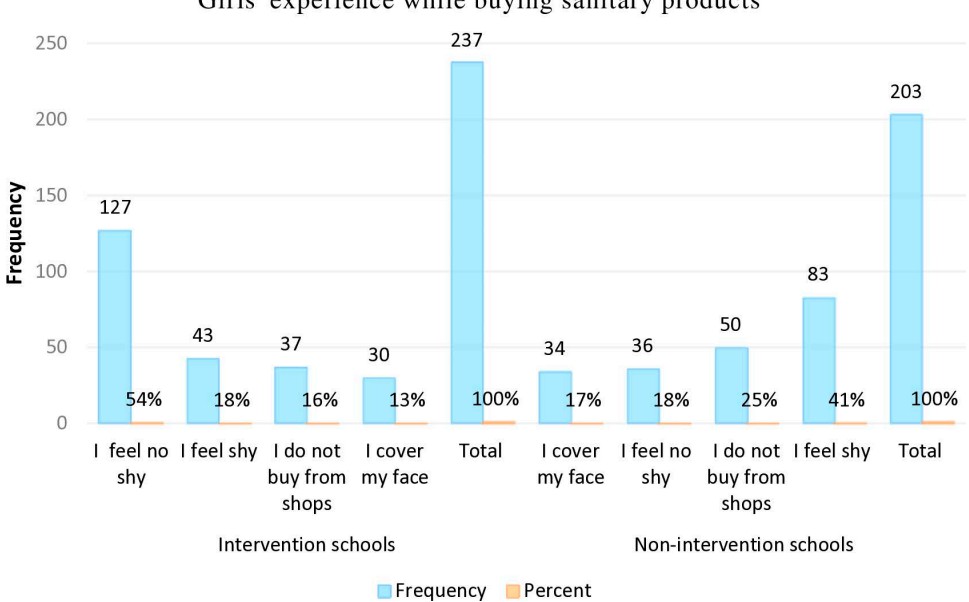

**Fig 1. Comparing girls' confidence while purchasing sanitary products.**

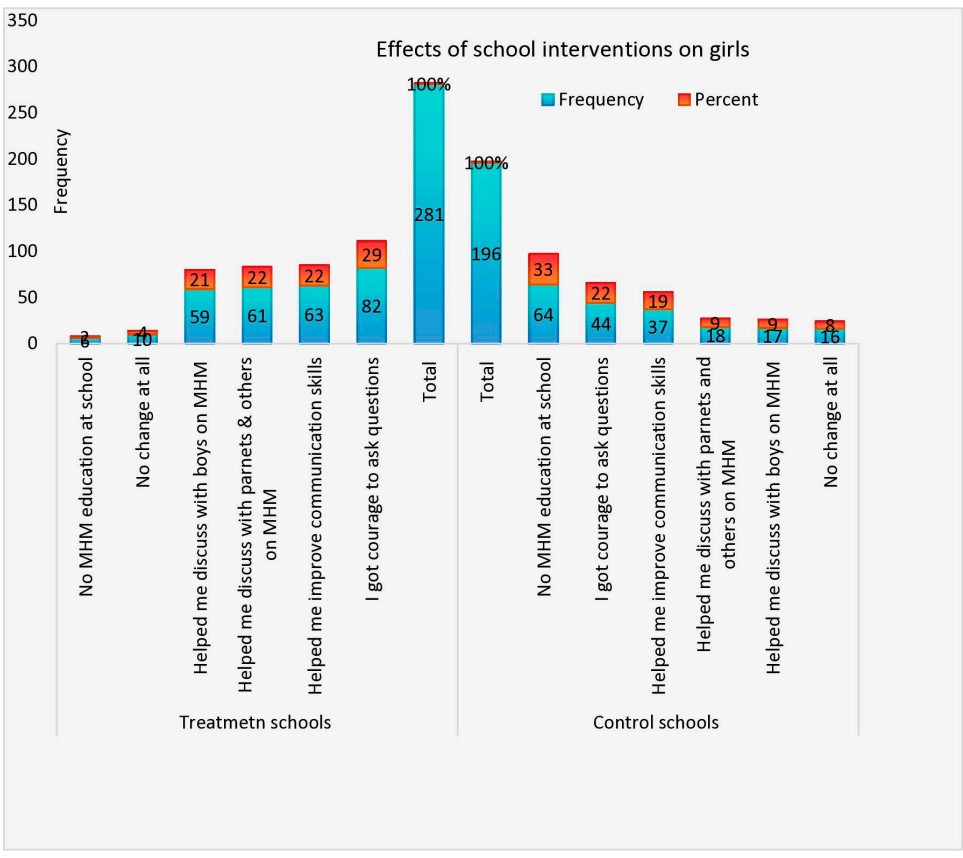

**Fig 2. Effects of school interventions on girls' social interaction.**

**Table 2. T-test results: Effects of school interventions on girls' social interactions.**

| Independent samples *t*-test results | | | | | | | | | | |
|---|---|---|---|---|---|---|---|---|---|---|
| | | Levene's test for equality of variances | | T-test for Equality of Means | | | | | | |
| | | *F* | *Sig.* | *t* | *df* | *Sig. (2-tailed)* | *Mean Difference* | *Std. Error Difference* | *95% Confidence Interval of the Difference* | |
| | | | | | | | | | *Lower* | *Upper* |
| I dare to ask questions in class | Equal variances assumed | 19.349 | 0.000 | 2.302 | 182 | 0.022 | 0.161 | 0.070 | 0.023 | 0.299 |
| | Equal variances not assumed | | | 2.319 | 181.924 | 0.022 | 0.161 | 0.069 | 0.024 | 0.298 |
| It helped me improve my communications skill | Equal variances assumed | 10.569 | 0.001 | 1.599 | 182 | 0.112 | 0.106 | 0.066 | −0.025 | 0.236 |
| | Equal variances not assumed. | | | 1.612 | 181.983 | 0.109 | 0.106 | 0.066 | −0.024 | 0.235 |
| It helped me discuss with boys on MHM | Equal variances assumed | 14.256 | 0.000 | 2.543 | 182 | 0.012 | 0.283 | 0.111 | 0.063 | 0.503 |
| | Equal variances not assumed | | | 2.684 | 117.885 | 0.008 | 0.283 | 0.105 | 0.074 | 0.492 |
| It helped me discuss with parents and others on MHM | Equal variances assumed | 63.424 | 0.000 | 3.573 | 182 | 0.000 | 0.212 | 0.059 | 0.095 | 0.329 |
| | Equal variances not assumed | | | 3.667 | 169.380 | 0.000 | 0.212 | 0.058 | 0.098 | 0.326 |
| No change at all | Equal variances assumed | 4.993 | 0.027 | −1.107 | 182 | 0.270 | −0.042 | 0.038 | −0.117 | 0.033 |
| | Equal variances not assumed | | | −1.088 | 157.158 | 0.278 | −0.042 | 0.039 | −0.118 | 0.034 |
| No MHM education at school | Equal variances assumed | 357.730 | 0.000 | −6.501 | 182 | 0.000 | −0.341 | 0.053 | −0.445 | −0.238 |
| | Equal variances not assumed | | | −6.179 | 103.847 | 0.000 | −0.341 | 0.055 | −0.451 | −0.232 |

2024 were analysed to assess the effects of unsafe MHM practices on girls' health. During this period, 41 females sought treatment for illnesses linked to inadequate MHM practices, 14 of whom (34%) were menstruating students. Clinical evidence identified excessive sanitary pad use and limited knowledge of proper MHM practices as the primary causes of these health problems.

However, quantitative data revealed that 43.8% of girls in intervention schools practised genital hygiene three times a day during menstruation, compared to only 19% in control schools. Furthermore, 51.1% of girls in intervention schools reported cleaning their genitals three to four times daily, whereas only 33.3% of girls in control schools followed this practice (Fig 3).

Regarding the emotional and social aspects of girls' health, 58% of female students in control schools reported feeling anxious or fearful when purchasing sanitary products, whereas only 31% of students in intervention schools experienced the same (Fig 1). Additionally, 21% of girls in intervention schools stated that MHM education empowered them to discuss menstruation openly with boys, parents, and others, compared to just 9% in control schools (Fig 2).

Data from the health centre highlighted waterborne diseases as a major challenge for communities, particularly affecting female students. During the reporting period, the centre treated 125 female patients, 51 of whom (41%) were students. The most prevalent waterborne

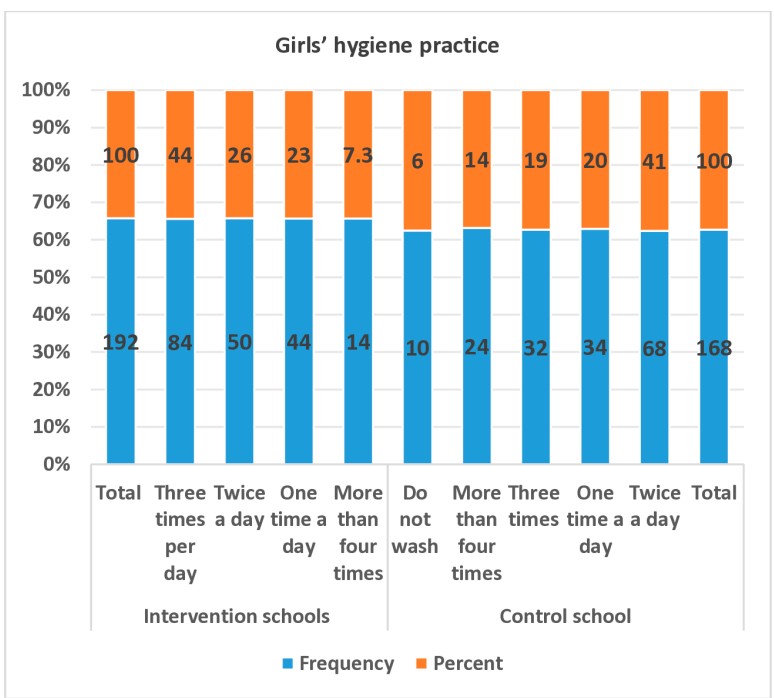

**Fig 3. Girls' genital hygiene practice during menstruation.**

illness was diarrhoea, primarily caused by inadequate access to clean water and water contamination. This finding aligned with data from the district education office, which reported that 70% of schools in the district lacked access to improved water sources.

Observations of school facilities and hygiene services, combined with feedback from focus group discussions, indicated that two of the three intervention schools had access to improved water sources. In contrast, none of the control schools had such access.

Data analysis revealed that when water was available at schools, girls utilized it for MHM practices. Specifically, 41% of girls in intervention schools used school water to wash sanitary pads, and 17% for genital hygiene, compared to 24% and 8%, respectively, in control schools (Fig 4).

The Independent Samples T-test for water use showed that, except for one case, all comparisons had p-values ranging from 0.001 to 0.022, both below the significance level of 0.05. Thus, $P < \alpha$ (0.001 < 0.05 and 0.022 < 0.05) (Table 3).

## 3.4. Effects of school interventions on girls' class attendance

**3.4.1. Qualitative results.** Qualitative interviews revealed that menstruating girls and their mothers identified family responsibilities, such as caring for a sick relative, participating in household chores related to ceremonies, and personal illness, as the primary reasons for class absenteeism. However, neither mothers nor girls mentioned menstruation as a factor for missing classes.

A female teacher serving as an MHM facilitator at an intervention school stated, "Menstruating students do not miss classes during their period because they receive better hygiene services at school than at home." In contrast, teachers from control schools reported that menstruating girls frequently missed classes due to a lack of MHM services. Additionally, girls often hesitated to disclose the reasons for their absenteeism due to fear of stigma and judgement.

### Benefits of school water to menstruating girls

**Fig 4. Benefits of school water for girls' MHM practice.**

**Table 3. T-test results: Multiple benefits of school water to menstruating girls.**

**Independent Samples Test results**

| | | Levene's Test for Equality of Variances | | t-test for Equality of Means | | | | | | |
|---|---|---|---|---|---|---|---|---|---|---|
| | | F | Sig. | t | df | Sig. (2-tailed) | Mean Difference | Std. Error Difference | 95% Confidence Interval of the Difference | |
| | | | | | | | | | Lower | Upper |
| Menstrual pad washing | Equal variances assumed | 5.779 | 0.017 | 5.668 | 182 | 0.001 | 0.386 | 0.068 | 0.252 | 0.52 |
| | Equal variances not assumed | | | 5.639 | 174.45 | 0.001 | 0.386 | 0.068 | 0.251 | 0.521 |
| Genital cleaning | Equal variances assumed | 48.105 | 0 | 3.183 | 182 | 0.002 | 0.19 | 0.06 | 0.072 | 0.308 |
| | Equal variances not assumed | | | 3.256 | 173.37 | 0.001 | 0.19 | 0.058 | 0.075 | 0.305 |
| Drinking and hand washing | Equal variances assumed | 48.809 | 0 | 3.493 | 182 | 0.001 | 0.232 | 0.067 | 0.101 | 0.364 |
| | Equal variances not assumed | | | 3.547 | 180.06 | 0.001 | 0.232 | 0.065 | 0.103 | 0.362 |
| Cleaning rooms | Equal variances assumed | 5.106 | 0.025 | 1.117 | 182 | 0.266 | 0.074 | 0.066 | −0.056 | 0.204 |
| | Equal variances not assumed | | | 1.123 | 181.63 | 0.263 | 0.074 | 0.065 | −0.056 | 0.203 |
| No water at school | Equal variances assumed | 407.451 | 0 | −9.438 | 182 | 0.001 | −0.517 | 0.055 | −0.625 | −0.409 |
| | Equal variances not assumed. | | | −8.999 | 108.45 | 0.001 | −0.517 | 0.057 | −0.631 | −0.403 |

**3.4.2. Quantitative results.** The quantitative analysis showed that 68.3% of girls in intervention schools attended classes during menstruation, compared to only 29.8% in control schools (Fig 5).

Among girls in control schools, the top three reported reasons for absenteeism during menstruation were lack of sanitary pads (31%), lack of water (17%), and menstrual pain or discomfort related to toilet facilities (11%). In contrast, girls in intervention schools cited menstrual pain (9%), lack of water and sanitary pads (7.8%), and inadequate toilet facilities (6.4%) as their primary reasons for absenteeism. Additionally, about 70.2% of girls in control schools reported missing classes due to MHM challenges, compared to only 31.7% in intervention schools (Fig 5).

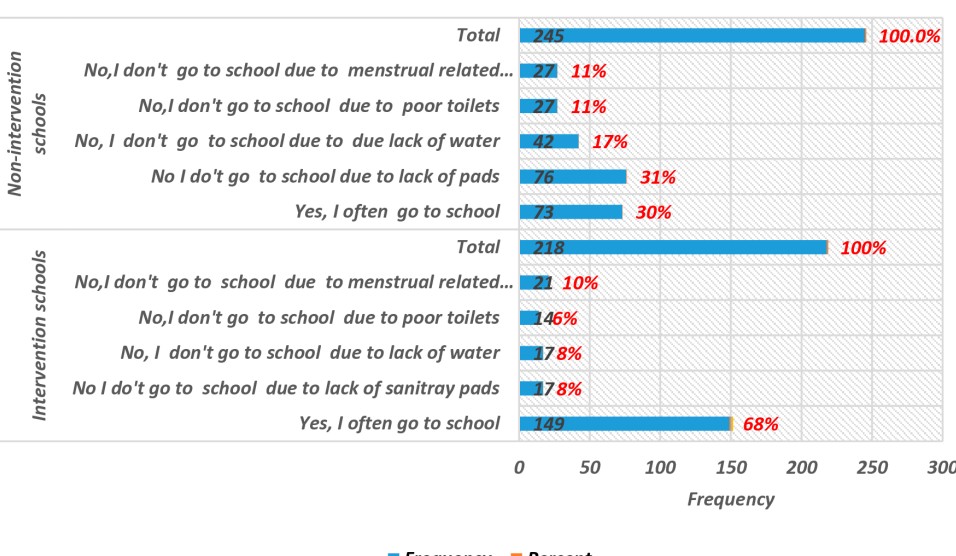

*Girls class attendance during menstruation*

**Fig 5. Girls' class attendance during menstruation.**

The independent samples t-test analysing differences in class absenteeism between girls in intervention and control schools showed that nearly all characteristics had p-values ranging from 0.001 to 0.004, which are below the significance level of 0.05 (Table 4). Therefore, P <α (0.001 < 0.05; 0.004 < 0.05).

Additionally, girls were asked to evaluate the reasons for their class absenteeism during both menstrual and non-menstrual periods. In control schools, the most frequently reported reason was family responsibilities (38%), followed by a lack of sanitary pads (26%) and menstrual pain or other illnesses (18%) (Fig 6).

In contrast, girls in intervention schools ranked menstrual pain as the leading cause of absenteeism (31%), closely followed by family responsibilities (30%) and other illnesses (21%) (Fig 6).

### 3.5. Effects of school interventions on academic performance

**3.5.1. Girls' study time.** The study revealed that 77.6% of girls in intervention schools reported having adequate time to study at home, compared to only 41.1% in control schools. Furthermore, 53% of girls in control schools cited family responsibilities as a barrier to study time, whereas only 10.9% of girls in intervention schools mentioned this as a factor (Fig 7).

The P-value for the difference in study time between the two groups of girls is 0.011, which is below the significance level. Thus, P <α, with 0.011 < 0.05 (Table 5).

**3.5.2. Academic performance.** To compare the educational performance of girls in intervention and control schools, the average scores for each group were calculated over a six-month period. In intervention schools, 333 girls (including both menstruating and non-menstruating) were assessed. The results showed that 2% of girls scored "excellent," 4% scored "very good," 14% scored "good," and 29% scored "fair" (Fig 8).

In contrast, 293 girls from control schools were assessed, yielding the following results: 3% scored "excellent," 9% scored "very good," 16% scored "good," and 25% scored "fair." These ratings are based on the standards established by the Ethiopian Ministry of Education for primary

**Table 4. T-test results: Effects of school interventions on class attendance.**

| Independent sample test results | | | | | | | | | |
|---|---|---|---|---|---|---|---|---|---|
| | | Levene's Test for Equality of Variances | | t-test for equality of means | | | | | |
| How often do you go to school during menstruation? | | F | Sig. | t | df | Sig. (2-tailed) | Mean Difference | Std. Error Difference | 95% Confidence Interval of the Difference |
| | | | | | | | | | Lower / Upper |
| Yes, I often go to school | Equal variances assumed | 30.100 | 0.000 | 5.104 | 182 | 0.000 | 0.345 | 0.068 | 0.212 / 0.479 |
| | Equal variances not assumed | | | 5.048 | 167.030 | 0.000 | 0.345 | 0.068 | 0.210 / 0.480 |
| No, I do not go to school due to lack of sanitary pads | Equal variances assumed | 160.112 | 0.000 | −6.081 | 182 | 0.000 | −0.362 | 0.059 | −0.479 / −0.244 |
| | Equal variances not assumed | | | −5.886 | 132.432 | 0.000 | −0.362 | 0.061 | −0.483 / −0.240 |
| No, I do not go to school due to lack of water | Equal variances assumed | 40.802 | 0.000 | −3.022 | 182 | 0.003 | −0.164 | 0.054 | −0.271 / −0.057 |
| | Equal variances not assumed | | | −2.945 | 144.171 | 0.004 | −0.164 | 0.056 | −0.274 / −0.054 |
| No, I do not go to school due to poor toilets | Equal variances assumed | 20.063 | 0.000 | −2.164 | 182 | 0.032 | −0.103 | 0.048 | −0.197 / −0.009 |
| | Equal variances not assumed | | | −2.112 | 146.544 | 0.036 | −0.103 | 0.049 | −0.199 / −0.007 |
| No, I do not go to school, due to menstrual-related pain | Equal variances assumed | 3.996 | 0.047 | −0.996 | 182 | 0.321 | −0.051 | 0.051 | −0.151 / 0.050 |
| | Equal variances not assumed | | | −0.985 | 168.278 | 0.326 | −0.051 | 0.051 | −0.152 / 0.051 |

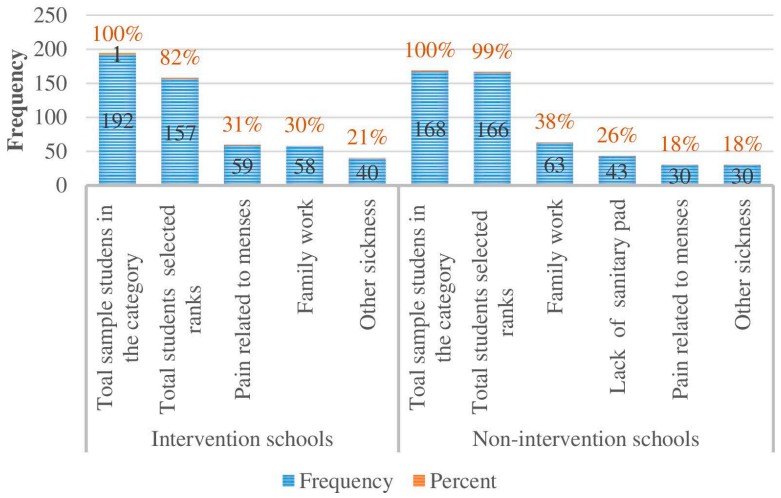

**Fig 6. Major factors of girls' class absenteeism - menstrual and non-menstrual periods.**

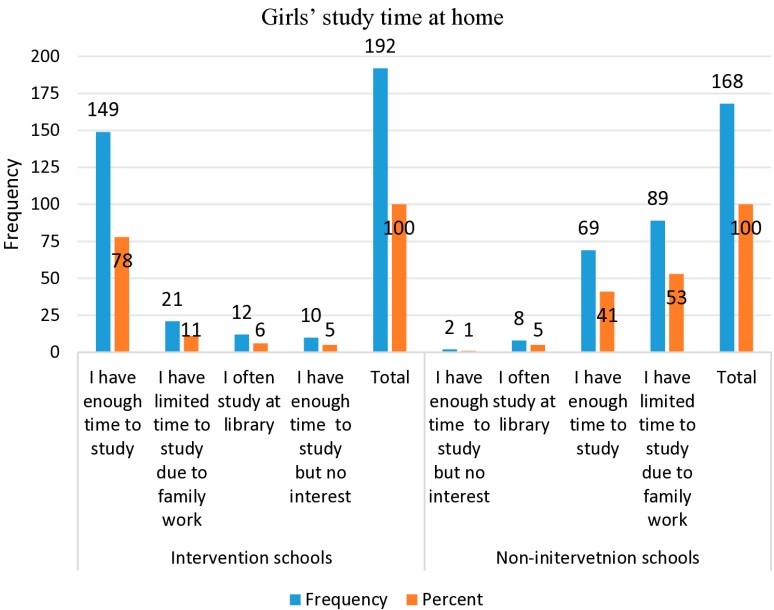

**Fig 7. Girls' study time at home.**

**Table 5. T-test results on girls' study time.**

| Independent Samples Test | | | | | | | | | | |
|---|---|---|---|---|---|---|---|---|---|---|
| | | Levene's Test for Equality of Variances | | t-test for Equality of Means | | | | | | |
| | | F | Sig. | t | df | Sig. (2-tailed) | Mean Difference | Std. Error Difference | 95% Confidence Interval | |
| | | | | | | | | | Lower | Upper |
| Do you have enough time to study at home? | Equal variances assumed | 0.846 | 0.358 | -3.522 | 358 | 0.011 | -0.295 | 0.084 | -0.460 | -0.130 |
| | Equal variances not assumed | | | -3.560 | 357.740 | 0.010 | -0.295 | 0.083 | -0.459 | -0.132 |

and secondary students. Notably, a similar proportion of girls from both groups received "good" or "fair" ratings: 43% in intervention schools and 41% in control schools (Fig 8).

## 4. Discussion

The demographic analysis indicated that the majority of participants were girls aged 13–15 years, comprising 69% of the sample. This was followed by girls aged 16–18, who made up 16%, and those aged 10–12, accounting for 15%. Ethnically, 98% of the participants identified as Amhara, with smaller percentages identifying as Agew (1%) and Tigre (0.8%). Regarding religion, 84% of the girls identified as Orthodox, 9% as Protestant, and 6% as Muslim.

The analysis of parental education levels revealed low literacy rates, with 66% of fathers able to read and write, compared to 52% of mothers. Household income was predominantly derived from agriculture (82%), while a smaller portion (8.3%) relied on business ventures (Table 1).

**Empowering girls through school interventions** Access to adequate MHM knowledge, skills, and Water, Sanitation, and Hygiene (WASH) services plays a critical role in enhancing

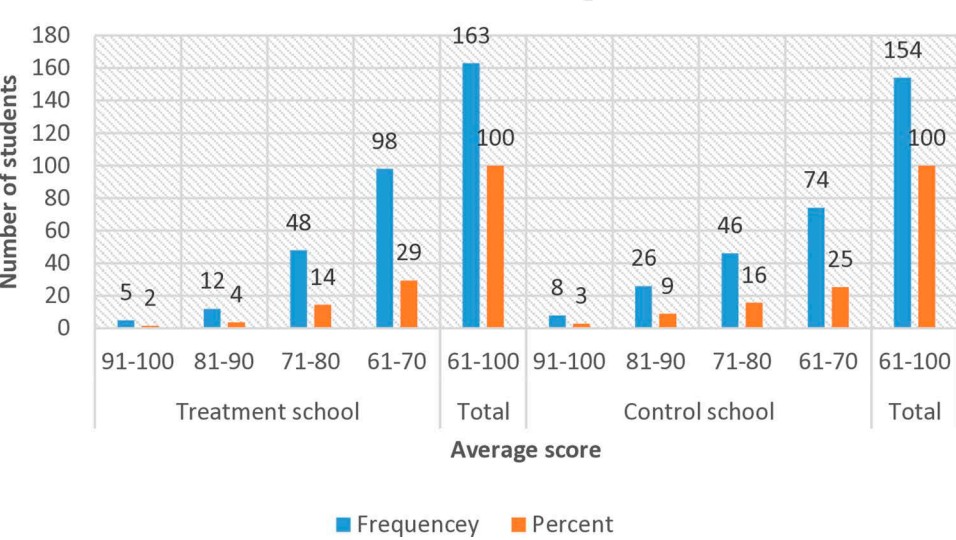

**Fig 8. First-semester academic performance.**

the confidence and empowerment of menstruating girls. Qualitative findings from focus group discussions with mothers, menstruating girls, and teachers revealed that girls in intervention schools were more confident discussing MHM with their peers, teachers, and mothers compared to those in control schools.

Quantitative data further reinforced this trend, with 54% of girls in intervention schools reporting confidence in purchasing sanitary products, compared to just 18% in control schools. In contrast, 41% of girls in control schools expressed fear when buying sanitary products, while only 18% of those in intervention schools felt the same (Fig 1).

In terms of social interactions, 21% of girls in intervention schools engaged in MHM discussions with their peers, including boys, and 22% discussed it with their parents. This was in stark contrast to only 9% of girls in control schools participating in such conversations (Fig 2).

An Independent Samples T-test further analysed these differences, confirming statistical significance with a P-value of 0.001. Since P <α (0.001 < 0.05), the results indicate a significant difference in the empowerment of menstruating girls between intervention and control schools (Table 2).

The significant difference in girls' empowerment between intervention and control schools can be largely attributed to disparities in access to MHM education and school WASH services. The school interventions not only enhanced girls' MHM knowledge and skills but also provided platforms for open discussions with peers, teachers, and parents (S1 Fig). These discussions, coupled with improved access to resources, helped increase their confidence, particularly in purchasing sanitary products. Moreover, the enhanced WASH services in schools allowed girls to practice safe MHM, alleviating concerns about potential bloodstains and enabling them to focus more on their education (S1 Table).

The findings of the study align with an assessment report by UNICEF, Plan International, Save the Children, and Emory University, which identified WASH facilities, MHM education, practical guidance, and access to sanitary products as critical components for girls' empowerment. The report further highlighted that the lack of these services can lead to unsafe MHM practices, anxiety about staining clothes, and increased teasing from boys [10].

Furthermore, reports by UNICEF and Columbia University over the past decade have underscored the importance of school WASH interventions in empowering girls' education [11]. Additionally, studies in Niger and Uganda on MHM and school attendance have demonstrated that improved school infrastructure, MHM education, and access to WASH services significantly enhance girls' knowledge and practices related to MHM, leading to positive outcomes in both health and education [5,27].

**Effects of school interventions on girls' physical, emotional, and social health** The absence of school-based MHM education and inadequate WASH services exposes girls to unsafe hygiene practices, leading to confusion, anxiety, and negative impacts on their physical, emotional, and social well-being [10].

Health centre data from September 2022 to February 2024, focusing on MHM and waterborne diseases, revealed that limited MHM knowledge and excessive use of sanitary products contributed to genital bruising and related discomfort among women and female students. During this period, menstruating girls accounted for 34% of the health centre's patients.

However, school-based MHM education and WASH interventions significantly improved girls' MHM knowledge and practices, leading to positive health outcomes. The study found that 44% of girls in intervention schools practiced genital hygiene three times a day during menstruation, compared to only 19% in control schools. Additionally, 51% of girls in intervention schools maintained a hygiene routine of three to four times daily, whereas only 33% of girls in control schools did the same (Fig 3).

Furthermore, while 58% of girls in control schools reported feeling shy or fearful when purchasing sanitary products, this figure dropped to 31% among girls in intervention schools. These results suggest that the school interventions significantly reduced these fears, contributing to improved emotional health among girls in intervention schools (Fig 1).

Additionally, 21% of girls in intervention schools felt comfortable discussing MHM with boys, and 22% with their parents, compared to only 9% in control schools. The findings point to enhanced social well-being and communication among girls in intervention schools (Fig 2).The interventions also led to improved water access in most intervention schools, enabling girls to practice safe menstrual hygiene management. For instance, 41% of girls in intervention schools used water for washing sanitary pads, and 17% for genital hygiene, compared to just 24% and 8% in control schools, respectively (Fig 4).

The disparity in water usage for safe MHM practices between intervention and control schools was statistically significant, with a P-value of 0.001, which is below the 0.05 threshold. Therefore, since P <α (0.001 < 0.05) (Table 3), there was a significant difference between the two groups in utilizing school water to support safe MHM practices.

These research findings align with studies from Bangladesh and Nepal, which reported that access to MHM education and WASH services reduce stress and discomfort among girls while improving their hygiene practices [3]. Similarly, a study in India found that access to WASH services and improved MHM practices helped mitigate negative health effects, such as infections related to reusable sanitary products [25].

However, the health centre report from the study district, which identified diarrhoea, as a WASH-related challenge affecting students, including girls, was not supported by either the qualitative or quantitative findings of this study. Female students ranked diarrhoea as the least significant cause of class absenteeism (Fig 6).

This discrepancy may be linked to cultural and educational factors. In the study district and surrounding region, the term "diarrhoea" carries a negative connotation, as the local term is often used as an insult. This cultural stigma may have made focus group participants reluctant to discuss the issue openly.

Another possible explanation is that mothers and girls may not distinguish diarrhoea as a specific health concern, instead referring to it more generally as "sickness." This could explain why girls categorized absenteeism due to illness under the broader category of "other sickness." In both intervention and control schools, girls ranked "other sickness" fourth, following family responsibilities, lack of sanitary pads, and menstrual hygiene-related pain among six possible reasons for absenteeism. As a result, diarrhoea may have been overlooked or grouped under this broader category (Fig 6).

Overall, the research findings highlight that school interventions can significantly enhance girls' emotional, social, and physical well-being, underscoring the importance of implementing MHM and WASH programs in schools.

**Effects of school interventions on class attendance** Qualitative findings indicate that family responsibilities, such as caregiving for sick relatives, girls' health conditions, and menstrual-related pain, were the primary reasons for class absenteeism rather than menstruation alone. However, teachers from both intervention and control schools attributed absenteeism more to the availability of MHM and WASH services at school.

A teacher facilitating an MHM club in an intervention school stated, "During menstruation, girls prefer coming to school because they receive better MHM services here than at home." Similarly, a school principal and mothers of girls from intervention schools acknowledged that access to MHM services at school helped prevent absenteeism during menstruation.

In contrast, mothers of girls from control schools believed that the lack of MHM services, such as water and sanitary products, contributed to absenteeism. Female students echoed this concern, stating, "Most girls do not change sanitary pads at school due to lack of access to water, sanitary pads, and girl-friendly toilets."

Additionally, a school principal and an MHM club facilitator from a control school explained, "Although girls do not explicitly say they are absent due to menstruation, we believe inadequate MHM services contribute to their absenteeism during this time."

The quantitative results further supported the qualitative findings, showing that 68.3% of girls in intervention schools attended classes during menstruation, compared to only 29.8% in control schools. This indicates that access to MHM services in intervention schools reduced absenteeism by more than half compared to control schools (Fig 5).

Furthermore, nearly 48% of girls in control schools reported missing classes due to a lack of sanitary products and water, whereas only 15.6% of girls in intervention schools cited the same reason (Fig 5).

Class absenteeism during menstruation was also influenced by social interactions, as teasing by boys and others contributed to girls' absences. In intervention schools, 21% of girls felt comfortable discussing MHM with boys, compared to only 9% in control schools. Additionally, 22% of girls in intervention schools believed MHM education enabled them to discuss menstruation with parents and others, whereas only 9% of girls in control schools shared this view (Fig 2).

The qualitative findings underscore the essential role of the school environment in supporting MHM practices, while the quantitative results provide measurable insights into their influence on attendance. An Independent Samples T-test further examined this difference, yielding a P-value of 0.001, well below the 0.05 threshold (Table 4). This statistically significant difference in attendance is largely attributed to access to MHM education and WASH services in intervention schools, which were absent in control schools.

These findings align with research from the Tigray Region of northern Ethiopia, which demonstrated improved MHM practices and increased class attendance among girls following the provision of MHM education, sanitary products, and underwear [8]. Similarly, a study

in Ghana reported higher-class attendance among girls after receiving sanitary products and MHM education [14]. Furthermore, research in Kenya found that access to school WASH services significantly reduced absenteeism among primary school students [19].

However, the findings contrast with those of two other studies. Research in Nepal, for instance, found no improvement in girls' attendance despite the provision of menstrual cups to address sanitary product shortages [9]. Likewise, a study in Malawi reported that variations in access to WASH services across schools did not significantly influence girls' absenteeism [18].

However, the differences in context may explain the contrasting results. The Nepal study focused solely on providing menstrual cups without incorporating other essential factors, such as MHM education and WASH services, which play a crucial role in shaping hygiene behaviour. In addition, the Nepal approach represented a single-factor intervention rather than a comprehensive strategy.

Similarly, the Malawi study did not emphasize behaviour change promotion related to school WASH services. Although some schools had WASH facilities as part of broader social services, these facilities were not designed to influence girls' hygiene practices.

The findings of this research, however, suggest that improving girls' MHM practices and class attendance requires holistic school-based interventions that address the specific needs of menstruating girls. Additionally, school WASH services should target specific behavioural changes, requiring active involvement from all stakeholders, including school community members.

Thus, the findings indicate that menstrual hygiene-related absenteeism can be reduced through comprehensive, targeted MHM education and WASH interventions. However, school-based efforts alone cannot eliminate absenteeism, as other contributing factors, such as household chores and menstrual-related pain, also play a role. Lastly, the effects of menstruation on class attendance should be examined in the context of school-based MHM and WASH services, rather than in isolation from these services.

**Effects of school interventions on academic performance** The discussion on girls' absenteeism suggests that school interventions can improve class attendance and reduce menstrual-related absences. Additionally, the study found that these interventions enhanced girls' study time at home. Specifically, 78% of girls in intervention schools reported having sufficient study time, compared to 41% in control schools. In contrast, 53% of girls in control schools cited household responsibilities as a barrier to studying, whereas only 11% of girls in intervention schools faced the same challenge (Fig 7).

The findings indicate that school interventions nearly doubled girls' study time at home and reduced their household burden by fivefold compared to control schools. This improvement was largely due to the empowerment provided by these interventions, which equipped girls with the confidence and communication skills to negotiate with their parents—an opportunity not available to girls in control schools.

Further analysis using the Independent Samples T-test yielded a P-value of 0.001, well below the 0.05 threshold (Table 5), indicating a statistically significant difference attributable to the interventions.

A key question that arises is whether the improved class attendance, increased study time, and other positive effects of the interventions ultimately led to better academic performance for girls in the intervention schools.

To evaluate academic performance, the study analysed half-year average scores, categorizing them as "excellent," "very good," "good," or "fair" based on school ratings. Among 333 girls in intervention schools, 2%, 4%, 14%, and 29% received these ratings, respectively. In comparison, among 293 girls in control schools, the corresponding percentages were 3%, 9%, 16%, and 25% (Fig 7).

Overall, 12% of girls in control schools received a "very good" or "excellent" rating, compared to only 6% in intervention schools—half the proportion observed in control schools. However, for lower ratings ("good" and "fair"), girls in intervention schools performed slightly better than their counterparts (43% versus 41%), though the difference was not statistically significant.

If first-semester average scores determined grade promotion, 49% (163 out of 333) of menstruating girls in intervention schools would advance, compared to 53% (154 out of 293) in control schools.

These findings suggest that high-performing students are more affected by class attendance and study time, whereas the impact on lower-performing students is minimal. This implies that factors beyond attendance and study time contribute to variations in academic performance. However, it is important to note that the comparison was based on half-year academic performance data, which may have influenced the results.

These findings contrast with previous research showing a positive correlation between class attendance and academic performance [20,21]. For example, one study on *Class Attendance and Learning Outcomes* found that attendance significantly improved performance among low-achieving students [22]. While this partially aligns with the performance of low-achieving girls in intervention schools, the difference was not statistically significant.

## 5. Conclusion

School-based MHM and WASH interventions provide multiple benefits for menstruating girls. They help build confidence in communicating with peers, teachers, and parents, enhance MHM skills, and improve girls' ability to negotiate for study time and reduce domestic workloads.

Additionally, these interventions support physical, emotional, and social well-being of girls by promoting safe MHM practices, emotional regulation, and positive social interactions. They also reduce class absenteeism by providing sanitation and hygiene facilities and promoting supportive school environments. However, domestic responsibilities and menstrual pain remain significant barriers to girls' class attendance.

Despite improvements in attendance, study time, and social interactions, the effect on academic performance appears minimal, suggesting that other factors influence learning outcomes.

The findings highlight the need for comprehensive, sustained interventions tailored to girls' diverse needs to enhance MHM practices, empowerment, health, and school attendance. Continued support from school programs is essential for the long-term adoption of MHM practices and resilience to related challenges.

This study contributes to closing knowledge gaps in MHM and absenteeism research, particularly regarding girls' empowerment, health, and academic performance, and serves as a foundation for future studies.

The findings carry important implications for policymakers and organizations focused on girls' education. Although Ethiopia has a supportive policy framework for school MHM and WASH interventions, limited budget allocation hampers their effective implementation. Therefore, regional and district WASH sector bureaus should prioritize and allocate adequate funds to enhance MHM and WASH services in schools.

Additionally, non-governmental organizations, as key contributors to school MHM and WASH programs, should strengthen their collaboration with government agencies, actively engage girls in the process, and develop comprehensive, sustainable school WASH initiatives.

## Supporting information

**S1 Fig.** *Girls' knowledge on menstruation.* Girls from intervention schools benefited from school peer-to-peer education and teachers who took MHM trainings. This helped girls from intervention school to demonstrate improved hygiene knowledge and practice compared to non-intervention schools who lucked these opportunities.
(DOCX)

**S2 Fig. Girls' confidence to request menstrual break during classes.** Menstruating girls from intervention school improved their social relations with boys and teachers due to the school intervention and this was reflected in their confidence to request their teachers, male and female, for menstrual break during classes, unlike the non-intervention school girls who were limited to female teachers.
(DOCX)

**S1Table. G***irls' experience to change sanitary pads at schools.* In non-intervention schools, girls lacked access to dedicated clubrooms where they could change sanitary pads, rest when feeling unwell and exchange hygiene education. This absence significantly impacted their comfort and support for managing menstrual health at school, leading to a much higher proportion of girls refraining from changing sanitary pads during school hours—44% compared to just 4.2% of the girls from intervention schools.
(DOCX)

## Acknowledgments

The authors acknowledge Ato Yalew Tizazu and Wondemu Sisay for their support in data collection. We also appreciate Lasta District Education Office for approving the research and teachers for facilitating the data collection process.

## Author contributions

**Conceptualization:** Fisseha Atale Andargie.

**Investigation:** Fisseha Atale Andargie.

**Methodology:** Fisseha Atale Andargie, Femi R. Tinuola.

**Project administration:** Fisseha Atale Andargie.

**Resources:** Fisseha Atale Andargie.

**Supervision:** Femi R. Tinuola.

**Validation:** Femi R. Tinuola.

**Writing – original draft:** Fisseha Atale Andargie.

**Writing – review & editing:** Femi R. Tinuola.

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
