## [Decision Letter · Decision Letter 0]

7 Jan 2025

PONE-D-24-55215Effects of school menstrual hygiene, water and sanitation interventions on girls’ empowerment, health and educational outcomes: Lasta District, Amhara Regional State, EthiopiaPLOS ONE

Dear Dr. Andargie,

Thank you for submitting your manuscript to PLOS ONE. After careful consideration, we feel that it has merit but does not fully meet PLOS ONE’s publication criteria as it currently stands. Therefore, we invite you to submit a revised version of the manuscript that addresses the points raised during the review process.

 All three reviewers have provided extensive comments to the writing of the manuscript which need to be addressed. Please submit your revised manuscript by Feb 21 2025 11:59PM. If you will need more time than this to complete your revisions, please reply to this message or contact the journal office at plosone@plos.org . Please include the following items when submitting your revised manuscript:

We look forward to receiving your revised manuscript.

Kind regards,

Alison Parker

Academic Editor

PLOS ONE

Journal Requirements:

3. Please ensure that you refer to Figure 8 in your text as, if accepted, production will need this reference to link the reader to the figure.

Reviewers' comments:

Reviewer's Responses to Questions

**Comments to the Author**

1. Is the manuscript technically sound, and do the data support the conclusions?

Reviewer #1: Partly

Reviewer #2: Yes

Reviewer #3: Yes

2. Has the statistical analysis been performed appropriately and rigorously? 

Reviewer #1: Yes

Reviewer #2: Yes

Reviewer #3: No

3. Have the authors made all data underlying the findings in their manuscript fully available?

Reviewer #1: Yes

Reviewer #2: Yes

Reviewer #3: Yes

4. Is the manuscript presented in an intelligible fashion and written in standard English?

Reviewer #1: Yes

Reviewer #2: Yes

Reviewer #3: Yes

5. Review Comments to the Author

Reviewer #1: The author needs to have all the grammatical errors corrected before submitting the paper for review. Some paragraphs are too wordy, this needs to be addressed without interfering with the meaning. There is poor flow of sentences as well, Some parafraphs do not trelate to the previous ones

Reviewer #2: I appreciate the work addressing this timely public health issue. I would like to raise few concerns regarding the study:

1. Method section

You mentioned that you utilized both methods. Could you clarify the purpose of the cross-sectional study? Additionally, what type of experimental design did you implement (e.g., true randomization, quasi-experiment, longitudinal, etc.)?

2. Details of the Intervention

The description of the intervention lacks specificity in terms of quantity, frequency, and duration. Additionally, the content of the intervention is described too superficially, making it difficult to derive meaningful insights or formulate policy recommendations based on the findings.

3. Contamination

The manuscript does not clearly outline the strategies implemented to prevent contamination or to buffer against potential biases.

4. Ethical Considerations

It is important to understand how the researchers addressed ethical dilemmas faced by non-intervention schools. Clarification on this matter is necessary.

Students under the age of 18 are included in your study. In this case, you must obtain both assent from the students and consent from their parents. Why was not assent considered?

5. Quality Control Mechanisms

What quality control mechanisms were in place to ensure the fidelity of the intervention, particularly regarding educational components? Blinding, randomization, and other quality control mechanisms need to be clearly articulated.

6. Mixed Methods Approach

The implementation of mixed methods is noted. Please clarify the sequence of the qualitative and quantitative studies: which method was conducted first, and what was the rationale behind this choice?

7. Controlling for Confounding Factor

Given that school performance can be influenced by multiple factors, it is essential to explain how the study controlled for the effects of other confounding variables on both school attendance and educational performance.

I recommend that all these concerns be addressed in the body of the manuscript to enhance its clarity and robustness.

Reviewer #3: Thank you for asking me to review this paper on “Effects of school menstrual hygiene, water, and sanitation interventions on girls’ empowerment, health, and educational outcomes: Lasta District, Amhara Regional State, Ethiopia." It is exciting research. However, the following issues must be improved before it is accepted for publication.

Title: The title of this study, which is stated as "Effects of School Menstrual Hygiene, Water, and Sanitation Interventions on Girls’ Empowerment, Health, and Educational Outcomes,” needs modifications. The intervention section, as indicated in lines 184–188, includes MHM education and MHM rooms, which are not included in this title. Therefore the title of this study should be modified to “Effects of School Menstrual Hygiene Management and Water, Sanitation and Hygiene (WASH) Service Interventions on Girls’ Empowerment, health and Educational Outcomes.”

Abstract sections: lines 31–33 should be consistent with the modified title. Please modify it as “This study investigated the effects of school menstrual hygiene management and water, sanitation, and hygiene (WASH) service interventions on girls’ empowerment, health, and educational outcomes. And also use effect or impacts consistently throughout the documents.

The introduction sections: The author’s well explained the context of the study. However, lines 157-160 need rearrangements. Line 157, “…..The significance of this study...” should come after line 160.

Design and setting: in line 173, the authors stated, “Six primary second-cycle schools were selected within the district” without describing first how many primary second-cycle schools are there in the district. The authors should also describe how six primary second-cycle schools were selected. Is randomization conducted? Or is there a baseline difference between exp’tal and controlled schools? Otherwise, it is difficult to conclude whether the effects after six years were due to interventions or results of their baseline difference.

Line 180: “10 primary second cycle schools or 6? This contradicts what was stated in line 173.

The informed consent section from line 234-262 needs modification. The informed voluntary consent for a minor (age < 18 years)/vulnerable individual should be signed by his/her legal competent representative (e.g., a parent/guardian). But in this study, even though around 85% of study participants were less than 16 years old (as indicated in table 1), informed voluntary consent taken from parent/guardian was not explained

Results section: Please rewrite from lines 284 to 288. For example, “Parental education levels were generally low...” Do not interpret the findings in the result section; keep it for the discussion section.

Please remove the gender variable from table 1.

The conclusion section is not well written. The authors should stick to the findings of the study during the conclusion. For instance, line 764-765, ”The interventions significantly boosted their confidence in discussing menstruation and lowered anxieties,” should be removed as not a tool that measures anxiety was indicated in the result.

Please rewrite the structured recommendation for this study and remove lines 770–778.

Finally, there are numerous editorial and grammatical errors that should be corrected before this manuscript is accepted. For instance, the author should use MHM consistently throughout the documents once he/she defined menstrual hygiene management as (MHM), unless it comes after full stop or at the beginning of the paragraph.

6. PLOS authors have the option to publish the peer review history of their article (what does this mean? ). If published, this will include your full peer review and any attached files.

**Do you want your identity to be public for this peer review?** For information about this choice, including consent withdrawal, please see our Privacy Policy .

Reviewer #1: No

Reviewer #2: **Yes: ** Balem Demtsu Betsu

Reviewer #3: **Yes: ** Abera Cheru

---

## [Author Response · Author response to Decision Letter 1]

15 Feb 2025

Response to the Editor: Full ethics statement is presented under methods, sub-section 2.4 ethical issues. The ethical letter was shared, together with the manuscript and it can be shared again if necessary. In addition, guiding templates for the title and the body have been used to design the manuscript; references reviewed for errors. This research was non-clinical and therefore, laboratory protocol is not applicable for this study. Fig 8 reviewed. Figures used in the manuscript are changed to tif file using PACE and uploaded separately from the manuscript. However, the manuscript maintains figure numbers and captions on separate pages where figures should be placed. Other details are indicated on the ' response to reviewers section.

2. Response to reviewer 1: The authors made extensive editing to improve word economy, clarity ,grammar error and coherence for the full manuscript.

Response to reviewer 2: The school MHM and WASH intervention began in July 2018; however, the study on the school intervention started in April 2023. Therefore, it was not possible to conduct a baseline survey, as time had already passed by the time the study began. Therefore, it was not possible to contrast baseline data with school interventions’ outcomes. Instead, the study used control schools that did not have school interventions to compare against the schools that received the WASH interventions. Consequently, longitudinal study design was not feasible due to the absence of baseline data and therefore, the study used cross-sectional design to measure the commutative effect of the school intervention by comparing with control schools. The study used quasi experimental design, details indicated under the following section.

For the detail see under methods, section, 2.2 Study design . Detail response for all question to reviewer 2 is indicated on ' response to reviewers' and manuscript.

Response to reviewer3: Improvement has been made to the title and abstract and so does on the rest of the of manuscript. However, the word’ services’ may not be necessary as the reference in the title is the school program/the intervention in order to realize the MHM services . We felt it is also better to use the abbreviation , WASH, in the body, rather than in the title. Therefore, the revised title reads as’’ Effects of School Menstrual Hygiene Management, Water, Sanitation and Hygiene Interventions on Girls’ Empowerment, Health and Educational Outcomes.’’ The detial response to reveiwer 3 is also indicated on ' response to reviewers' and manuscript.

---

## [Decision Letter · Decision Letter 1]

27 Feb 2025

PONE-D-24-55215R1Effects of school menstrual hygiene management, water, sanitation and hygiene interventions on girls’ empowerment, health and educational outcomes: Lasta District, Amhara Regional State, EthiopiaPLOS ONE

Dear Dr. Andargie,

Thank you for submitting your manuscript to PLOS ONE. After careful consideration, we feel that it has merit but does not fully meet PLOS ONE’s publication criteria as it currently stands. Therefore, we invite you to submit a revised version of the manuscript that addresses the points raised during the review process. Reviewers 1 and 3 have some remaining minor comments that need to be addressed. Please submit your revised manuscript by Apr 13 2025 11:59PM. If you will need more time than this to complete your revisions, please reply to this message or contact the journal office at plosone@plos.org . Please include the following items when submitting your revised manuscript:

We look forward to receiving your revised manuscript.

Kind regards,

Alison Parker

Academic Editor

PLOS ONE

Journal Requirements:

Reviewers' comments:

Reviewer's Responses to Questions

**Comments to the Author**

1. If the authors have adequately addressed your comments raised in a previous round of review and you feel that this manuscript is now acceptable for publication, you may indicate that here to bypass the “Comments to the Author” section, enter your conflict of interest statement in the “Confidential to Editor” section, and submit your "Accept" recommendation.

Reviewer #1: All comments have been addressed

Reviewer #2: All comments have been addressed

Reviewer #3: All comments have been addressed

2. Is the manuscript technically sound, and do the data support the conclusions?

Reviewer #1: Yes

Reviewer #2: Yes

Reviewer #3: Yes

3. Has the statistical analysis been performed appropriately and rigorously? 

Reviewer #1: Yes

Reviewer #2: (No Response)

Reviewer #3: Yes

4. Have the authors made all data underlying the findings in their manuscript fully available?

Reviewer #1: Yes

Reviewer #2: Yes

Reviewer #3: Yes

5. Is the manuscript presented in an intelligible fashion and written in standard English?

Reviewer #1: Yes

Reviewer #2: Yes

Reviewer #3: Yes

6. Review Comments to the Author

Reviewer #1: The authors have done a great job. They need to work on the new comments and the grammer to be able to submit a qualityy paper

Reviewer #2: (No Response)

Reviewer #3: The author's well-responded for almost all comments, but still, the authors should make the measurement measures of confounding variables precise "from lines 337-366". The author should also remove the female variable entirely from table 1 with its frequency and percent; it is clear on the title as the study was conducted on girls

7. PLOS authors have the option to publish the peer review history of their article (what does this mean? ). If published, this will include your full peer review and any attached files.

**Do you want your identity to be public for this peer review?** For information about this choice, including consent withdrawal, please see our Privacy Policy .

Reviewer #1: No

Reviewer #2: **Yes: ** Balem Demtsu Betsu

Reviewer #3: **Yes: ** Abera Cheru

---

## [Author Response · Author response to Decision Letter 2]

4 Mar 2025

Reviewer 1. The Authors have made extra effort to improve the quality of the article by improving the mechanics-including grammar.

Reviewer 3. Authors tried their best to address comments raised by the reviewer, especially on confounding variables.

---

## [Editor Report · Decision Letter 2]

5 Mar 2025

Effects of school menstrual hygiene management, water, sanitation and hygiene interventions on girls’ empowerment, health and educational outcomes: Lasta District, Amhara Regional State, Ethiopia

PONE-D-24-55215R2

Dear Dr. Andargie,

We’re pleased to inform you that your manuscript has been judged scientifically suitable for publication and will be formally accepted for publication once it meets all outstanding technical requirements.

Kind regards,

Alison Parker

Academic Editor

PLOS ONE
---

## [Editor Report · Acceptance letter]

PONE-D-24-55215R2

PLOS ONE

Dear Dr. Andargie,

I'm pleased to inform you that your manuscript has been deemed suitable for publication in PLOS ONE. Congratulations! Your manuscript is now being handed over to our production team.

Kind regards,

on behalf of

Dr. Alison Parker

Academic Editor

PLOS ONE